# Clinical Significance of Tumor Size in Gross Extrathyroidal Extension to Strap Muscles (T3b) in Papillary Thyroid Carcinoma: Comparison with T2

**DOI:** 10.3390/cancers14194615

**Published:** 2022-09-23

**Authors:** Joonseon Park, Il Ku Kang, Ja Seong Bae, Jeong Soo Kim, Kwangsoon Kim

**Affiliations:** Department of Surgery, College of Medicine, The Catholic University of Korea, Seoul 06591, Korea

**Keywords:** papillary thyroid carcinoma, extrathyroidal extension, disease-free survival, TNM staging

## Abstract

**Simple Summary:**

In the 8th edition of the AJCC/UICC TNM staging system, T3b is defined as gross extrathyroidal extension (gETE) invading only the strap muscles from a tumor of any size. However, defining T3b according to tumor size remains controversial. We created new T3b categories according to tumor size (T3b-1, tumor ≤ 2 cm in T3b; T3b-2, 2 cm < tumor ≤ 4 cm in T3b). There was a significant difference only in T3b-2 compared to T2. Novel methods of evaluating the effect of gETE according to tumor size should be considered in further revisions of the AJCC/UICC TNM staging system.

**Abstract:**

The purpose of the present study was to compare the risk of recurrence between T2 and T3b papillary thyroid carcinoma (PTC) and the effect of tumor size on survival in T3b disease. A total of 634 patients with PTC who underwent thyroid surgery at a single center were retrospectively analyzed. Clinicopathological characteristics were compared according to the T category in the TNM staging system, with T3b divided into T3b-1 (tumor size, ≤2 cm) and T3b-2 (tumor size, 2–4 cm). Disease-free survival (DFS) and recurrence risk were compared between T2, T3b, T3b-1, and T3b-2. Tumor size was significantly larger in T2 than in T3b. A significant difference in recurrence was observed between T2 and T3b-2 but not between T2 and T3b-1. T3b-2 was identified as a significant risk factor for PTC recurrence. A significant difference in the DFS curve was observed between T2 and T3b-2. However, no significant differences in survival were observed between T2 and T3b or T3b-1. These results indicate that the prognostic impact of T3b may vary depending on tumor size. Further studies are required to determine the need for T classifications that account for tumor size and gETE invasion of the strap muscles.

## 1. Introduction

Papillary thyroid cancer is the most common thyroid cancer accounting for approximately 95% of all cases of thyroid cancer in Korea and approximately 80–90% of thyroid cancer cases worldwide [1,2,3,4]. The American Joint Commission on Cancer/Union for International Cancer Control (AJCC/UICC) revised the TNM staging system for differentiated thyroid cancer (DTC) in the 8th edition published in 2016. The concept of minimal extrathyroidal extension (mETE) was removed from the T3 category, and the T3b category, which is confirmed intraoperatively, was defined as gross extrathyroidal extension (gETE) invading only the strap muscles (sternohyoid, sternothyroid, thyrohyoid, or omohyoid muscles) from a tumor of any size [5]. This staging system distinguished anterior and posterior extrathyroidal extensions (ETEs) unlike previous editions [6].

Many studies have investigated the impact of ETE, both mETE and gETE [7,8,9,10,11,12,13,14,15,16]. As mETE has no effect on prognosis, including recurrence and mortality, it was removed from the revised TNM staging system [5,7,8,9]. However, the clinical significance of T3b remains controversial. Several studies have demonstrated that T3b has a negative impact on disease-free survival (DFS) and disease-specific survival (DSS) in DTC [10,11,12]. Conversely, other studies have reported that T3b has no effect on long-term recurrence and survival [13,14,15,16,17,18]. One study concluded that T3b had no difference in DSS compared with T1, but only T3a had a worse effect [18].

Although many studies have focused on the effect of T3b itself, there are still few studies on the effect of tumor size in T3b. Several studies have evaluated the effect of primary tumor size on the impact of T3b [15,16,19]. T3b of any tumor size has been posited to have no effect on DSS [16], and conversely, T3b with larger primary tumor size has a worse prognosis than T3b with smaller tumor size [15,19]. These previous study cohorts have included T3b disease with a tumor size of ≤ 4 cm or 1–4 cm [15,19]. Advancing from previous studies, we compared the effect of T3b by dividing the tumor size into 2 cm and 4 cm according to the size criterion for T1 and T2.

The purpose of the present study was to compare clinicopathological characteristics and DFS between patients with T2 and T3b disease. In addition, we aimed to clarify the significance of T3b on the recurrence of PTC according to tumor size.

## 2. Materials and Methods

### 2.1. Patients

T2 and T3b PTC who underwent thyroid surgery from March 2009 to December 2017 at Seoul St. Mary’s Hospital (Seoul, Korea) were included. In total, 44 patients were excluded from the study analysis due to insufficient data, loss of follow-up, and distant metastasis in 22, 19, and 3 patients, respectively. The medical charts and pathologic reports of 634 patients were retrospectively reviewed and analyzed. The mean follow-up duration was 110.5 ± 28.1 months (range, 52–159 months). All information including tumor size, multifocality, bilaterality, lymphatic invasion, vascular invasion, BRAFV600E positivity, number of harvested lymph nodes, and metastatic lymph nodes were confirmed by the pathologic reports, and only T3b was confirmed intraoperatively by surgeons (Figure 1). There were 325 patients with T2 disease and 309 patients with T3b disease. The present study was conducted in accordance with the Declaration of Helsinki (as revised in 2013). This study was approved by the Institutional Review Board of Seoul St. Mary’s Hospital, Catholic University of Korea (IRB No: KC22RISI0467), which waived the requirement for informed consent due to the retrospective nature of this study.

### 2.2. TNM Classification

The tumor (T) category was classified according to the 8^th^ edition of the AJCC/UICC TNM staging system [5]. T2 disease was defined as a tumor of >2 cm but ≤4 cm in the greatest dimension limited to the thyroid. T3b disease was defined as gETE invading only the strap muscles (sternohyoid, sternothyroid, thyrohyoid, or omohyoid muscles) from a tumor of any size. In the present study, we classified T3b-1 and T3b-2 based on tumor size (T3b-1, tumor size ≤ 2 cm in T3b; T3b-2, tumor size, 2–4 cm in T3b).

### 2.3. Surgical Treatment

Surgical extent was basically determined according to the American Thyroid Association (ATA) management guidelines, but prophylactic central lymph node dissection (CLND) was performed in all patients (Lobectomy + CLND: 209 (33%); Total thyroidectomy + CLND: 286 (45.1%); modified radical neck dissection (mRND): 139 (21.9%), respectively). mRND was performed only when pathologic diagnosis of lateral cervical lymph node metastasis on preoperative biopsy. If the lateral cervical lymph node metastasis was suspected radiologically but a preoperative biopsy was not confirmed, metastasis was confirmed through a frozen-section biopsy during surgery, and the surgical extent was expanded to mRND if it was positive.

### 2.4. Follow-Up Assessments

The ATA management guidelines were followed for providing postoperative care and follow-up [20]. Every 3–6 months for the first year and annually thereafter, all patients had physical examinations, serum thyroid function tests, and measurement of thyroglobulin and anti-thyroglobulin antibody concentrations. Thyroid ultrasonography was performed once a year. Radioactive iodine (RAI) ablation was performed 8–12 weeks following total thyroidectomy. Whole-body scans were performed 5–7 days after RAI ablation. Patients with suspected recurrence underwent additional diagnostic imaging procedures, such as computed tomography, positron emission tomography/computed tomography, and/or RAI WBS, during follow-up evaluation to determine the exact location and size of recurrent lesions. Disease recurrence was confirmed by imaging and pathologic examinations of ultrasound-guided fine-needle aspiration or surgical biopsy specimens.

### 2.5. Statistical Analyses

Continuous variables are presented as means with standard deviations, and the Student’s *t*-test was used for comparative analysis. Categorical variables are reported as numbers with percentages, and Pearson’s chi-square test or Fisher’s exact test was used for comparative analysis. Univariate Cox regression analyses were performed to validate DFS predictors. Statistically significant variables in univariate analyses were analyzed by multivariate Cox proportional hazard model. Hazard ratios (HRs) with 95% confidence intervals (CIs) were calculated. DFS curves were compared using Kaplan–Meier survival analysis, with the log-rank test to calculate significant differences. *p*-values less than 0.05 were considered statistically significant. All statistical analyses were performed using Statistical Package for the Social Sciences (version 24.0; IBM Corp., Armonk, NY, USA).

## 3. Results

### 3.1. Comparison of Baseline Clinicopathological Characteristics between T2 and T3b Disease

Table 1 shows the comparison of clinicopathological characteristics between patients with T2 and T3b disease. Patients with T3b disease were significantly older than the patients with T2 disease (50.0 ± 13.6 years vs. 43.5 ± 14.6 years; *p* < 0.001). The extent of surgery was significantly more extensive in T3b than T2 (*p* < 0.001). The proportions of patients with multifocal disease, bilateral disease, and lymphatic invasion were significantly higher in T3b than T2 (*p* = 0.001, *p* < 0.001, and *p* < 0.001, respectively). However, tumor size was significantly larger in T2 than T3b (2.7 ± 0.5 cm vs. 1.8 ± 1.0 cm; *p* < 0.001). N stage and TNM stage were significantly more advanced in T3b than in T2 (*p* < 0.001 for both). However, there was no statistically significant difference in the rate of recurrence between T2 and T3b (4.9% vs. 5.8%; *p* = 0.614).

### 3.2. Univariate and Multivariate Analyses of Recurrence Risk Factors in Patients with T2 and T3b

Table 2 displays the univariate and multivariate Cox regression analyses for identifying risk factors for recurrence of PTC in patients with T2 and T3b disease. Age, male sex, tumor size, lymphatic invasion, and N stage were significant risk factors in univariate analysis. Age (HR, 0.968; 95% CI, 0.944–0.993; *p* = 0.012), male sex (HR, 2.506; 95% CI, 1.277–4.918; *p* = 0.008), tumor size (HR, 1.434; 95% CI, 1.036–1.984; *p* = 0.030), and lymphatic invasion (HR, 4.822; 95% CI, 1.851–12.562; *p* = 0.001) were significant risk factors for recurrence of PTC in multivariate analysis. However, T3b was not identified as a risk factor for recurrence (HR, 1.185; 95% CI, 0.604–2.324; *p* = 0.621). The original table with all information including non-significant variables is presented in the Appendix A. Kaplan–Meier survival analysis demonstrated no significant difference in DFS between T2 and T3b disease (*p* = 0.620 by log-rank test; Figure 2).

### 3.3. Comparison of Baseline Clinicopathological Characteristics between T2, T3b-1, and T3b-2 Disease

The baseline clinicopathological characteristics of patients with T2, T3b-1, and T3b-2 disease are presented in Table 3. There were 217 patients with T3b-1 and 79 patients with T3b-2 disease. Patients with T3b-1 and T3b-2 disease had older age (*p* < 0.001); more extensive surgery (*p* < 0.001); and higher rates of multifocality (*p =* 0.002, *p =* 0.010, respectively), bilaterality (*p* < 0.001), and lymphatic invasion (*p* = 0.017, *p* < 0.001, respectively) compared to patients with T2 disease. Both N stage and TNM stage were significantly more advanced in patients with T3b-1 and T3b-2 disease than patients with T2 disease (T3b-1, *p* = 0.002, *p* < 0.001; T3b-2, *p <* 0.001, and *p* < 0.001, respectively). However, no significant difference in the rate of recurrence was observed between T2 and T3b-1 (4.9% vs. 2.8%; *p* = 0.212). Conversely, a significantly higher rate of recurrence was observed in patients with T3b-2 disease compared to patients with T2 disease (12.7% vs. 4.9%; *p* = 0.012).

### 3.4. Univariate and Multivariate Analyses of Risk Factors for Recurrence in T2 or T3b-1 Disease

Age was a significant risk factor for the recurrence of PTC in multivariate analysis (HR, 0.955; 95% CI, 0.923–0.988; *p* = 0.008). There was no significant difference in the risk of recurrence between T3b-1 and T2 disease (HR, 0.556; 95% CI, 0.218–1.422; *p* = 0.221; Table 4). The original table with all information including non-significant variables is presented in the Appendix A. There was no significant difference in DFS between T3b-1 and T2 disease in Kaplan–Meier survival analysis (*p* = 0.214 by log-rank test; Figure 3).

### 3.5. Univariate and Multivariate Analyses of Risk Factors for Recurrence in T2 and T3b-2 Disease

In multivariate analysis, age (HR, 0.959; 95% CI, 0.932–0.987; *p* = 0.004) and lymphatic invasion (HR, 3.208; 95% CI, 1.159–8.882; *p* = 0.025) were significant risk factors for recurrence of PTC (Table 5). T3b-2 was a significant risk factor for recurrence of PTC (HR 1.875; 95% CI, 0.798–4.403; *p* = 0.021). The original table with all information including non-significant variables is presented in the Appendix A. Significantly lower DFS was observed in patients with T3b-2 compared to patients with T2 disease in Kaplan–Meier survival analysis (*p* = 0.012 by log-rank test; Figure 3).

## 4. Discussion

In the present study, there was no difference in the risk of disease recurrence between T3b with smaller tumor size (≤ 2 cm) and T2 disease. Only T3b with larger tumor size (2–4cm) was associated with significantly increased risk of recurrence and decreased DFS compared to T2 disease. These results indicate that the effect of T3b disease varies depending on tumor size.

The most significant changes in the AJCC/UICC TNM staging system after revision in 2016 were in cut-off age, the T3 category, extent of N1a, and TNM staging. In particular, the T category was adjusted to be more concise; however, the T3b category remains controversial. Although T1, T2, and T3a disease are now classified according to tumor size, T3b is defined as gETE limited to the strap muscles “regardless of the tumor size.” Accordingly, the effect of tumor size in this category remains unclear.

We evaluated the effect of T3b according to primary tumor size. First, the risk of recurrence among all patients with T3b did not significantly differ from patients with T2 disease. This result is consistent with many previous studies that have compared T3b with T2 disease [13,14,15,16,17]. Song et al. reported no difference in DSS between patients with gETE and patients without gETE for PTC of 1–4 cm diameter [19]. In addition, DSS in patients with T3b disease did not differ from patients with T2 disease (HR 0.81; 95% CI 0.24–2.77; *p* = 0.737) in their study [15,19]. Tam et al. also concluded that there were no significant differences in DSS between T2 and T3b (*p* = 0.358) [17]. However, several previous studies have reported that T3b has a negative effect on oncologic outcomes [10,11,12]. These divergent results may be due to the effect of other variables on the impact of T3b disease. Several studies have examined the correlation between gETE and tumor size, reporting that larger tumors were more likely to have gETE invading the strap muscles [10,14,21,22]. Although many studies have demonstrated that T3b is closely associated with risk factors for recurrence, few studies have evaluated the effect of T3b disease on prognosis depending on tumor size. Liu et al. evaluated the T3b category by dividing patients into two groups with a tumor size cut-off of 1 cm and concluded that only T3b with tumor size greater than 1 cm had an adverse effect on prognosis in PTC [23]. Most previous studies have used tumor size cut-offs of either 1 cm or 4 cm in their inclusion criteria [10,14,21,22]. We attempted to evaluate the effect of tumor size in T3b disease using tumor size cut-offs of 2 cm and 4 cm, which are the size criteria for T1 and T2 disease.

The effect of T3b differed depending on tumor size. DFS was significantly worse in patients with T3b disease with larger tumor size (2–4cm) compared to patients with T2 disease; however, there was no difference in DFS between patients with T3b disease with smaller tumor size (≤ 2 cm) and patients with T2 disease. This result indicates that T3b with smaller tumor size (≤ 2 cm) may still be upstaged.

The multivariate analysis and DFS curves in the present study both demonstrated that patients with T3b disease with larger tumor size (2–4 cm) had a significantly increased risk for recurrence compared to patients with T2 disease. As the size category was the same in the comparison of T2 disease with T3b disease with larger tumor size (2–4 cm), the effect of strap muscle invasion could be accurately compared between the two groups. Therefore, we verified that the impact of T3b on the recurrence of PTC is dependent on the size of the primary cancer.

In the present study, we performed an analysis using new T categories. Using the concept of matching T3b to the existing size criteria used for T1 and T2 disease, we developed two new categories, T3b-1 and T3b-2, according to tumor size. We compared clinicopathological characteristics and oncologic outcomes between patients with T2 disease and patients with T3b-1 or T3b-2 disease. We observed no difference in DFS between patients with T2 and T3b-1 disease. In Addition, there was no significant difference in the simple sub-analysis comparing the recurrence rate and risk of T1 and T3b (Appendix A). It suggests that it may not be appropriate to classify T3b-1 as an aggressive stage. However, patients with T3b-2 disease had a higher risk of recurrence than patients with T2 disease. Therefore, we proposed a novel staging system of T2a—tumor size of 2–4 cm without gETE—and T2b—tumor size of 2–4 cm with gETE into the strap muscles. By maintaining the current T category size criteria and adding the concept of gETE into the strap muscles, T2a and T2b categories can be created by adding “a” if there is no gETE and “b” if gETE in the strap muscles is present. 

Recently, several studies have proposed modifications to the TNM staging criteria. Previous studies have concluded T3b does not affect prognosis, indicating the T3b category represents overstaging and should be considered for revision [15,16,17,24]. Song et al. presented a modified T classification that downgrades patients with T3b (≤4 cm) disease to the T2 stage to improve prognostication [15]. Yoon et al. proposed a modified TNM staging schema that removed strap muscle invasion to better reflect DSS and avoid overstaging of tumors [16]. However, in the present study, we concluded that T3b should be subdivided according to tumor size as the effect of T3b differs according to tumor size.

In the present study, younger age was demonstrated as a significant risk factor for recurrence in multivariate analyses of all subgroups, corroborating the results of previous studies [25,26,27]. In addition, previous studies have demonstrated that younger age is associated with metastatic lymph node involvement [28,29,30,31,32]. These studies support the need for optimal surgical methods, such as total thyroidectomy or central lymph node dissection, for PTC in younger patients due to more aggressive characteristics and a higher risk of recurrence [29].

In addition, the proportion of patients with BRAF^V600E^ mutation was significantly higher in the T3b, T3b-1, and T3b-2 groups compared to the T2 group. This suggests that BRAF mutations may contribute to the development of more advanced disease, such as cancer invasion. BRAF^V600E^ mutation is the most frequent genetic event in PTC and is found in approximately 80% of patients with PTC in Korea; however, the prevalence and patterns of BRAF mutations differ worldwide [33,34]. In our entire cohort, the proportion of patients with BRAF^V600E^ was 77.8%, which is similar to previous studies [33,35]. BRAF^V600E^ mutation is known to be associated with larger tumors and higher rates of gETE rates [34]. In addition, BRAF^V600E^ mutation is a poor prognostic factor and is associated with increased mortality [36]. In the present study, the proportions of patients with BRAF mutations in T2, T3b, T3b-1, and T3b-2 groups were 71.8%, 86.1%, 84.8%, and 88.9%, respectively. This result implies that the presence of the BRAF^V600E^ mutation may be associated with large tumor size and strap muscle invasion, as reported in previous studies. Our results demonstrate that the BRAF^V600E^ mutation is associated with aggressive tumor features but has no effect on the risk of disease recurrence.

The present study has several limitations. First, this was a retrospective study conducted at a single institution. Second, information including PTC subtypes / variants and radioiodine therapy were not included. As aggressive variants are associated with poor prognosis in PTC and radioiodine therapy might affect the local recurrence, further studies including these variables may verify the effect of T3b on survival more precisely. Lastly, there was no detailed subgroup analysis comparing T3b disease with T1 or T3a disease. Further studies should compare T1 disease with T3b disease with tumor size ≤ 2 cm and T3a disease with T3b disease with tumor size of > 4 cm. In addition, as AJCC/UICC TNM stages are determined by evaluating survival rather than recurrence, and overall survival and DSS should also be evaluated in future studies.

## 5. Conclusions

The impact of T3b may vary depending on tumor size. T3b with larger tumor size is a significant risk factor for recurrence in PTC, unlike T3b with a smaller tumor size. T3b with a smaller tumor size may still be upstaged due to a similar risk of recurrence compared to the T2 category. We propose a novel staging system of T2a as a tumor size of 2–4 cm without gETE, and T2b as tumor size of 2–4 cm with gETE into the strap muscles. Further studies may validate this novel classification system that adds the concept of gETE into the strap muscles to the tumor size criteria of the current TNM classification for PTC.

## Figures and Tables

**Figure 1 cancers-14-04615-f001:**
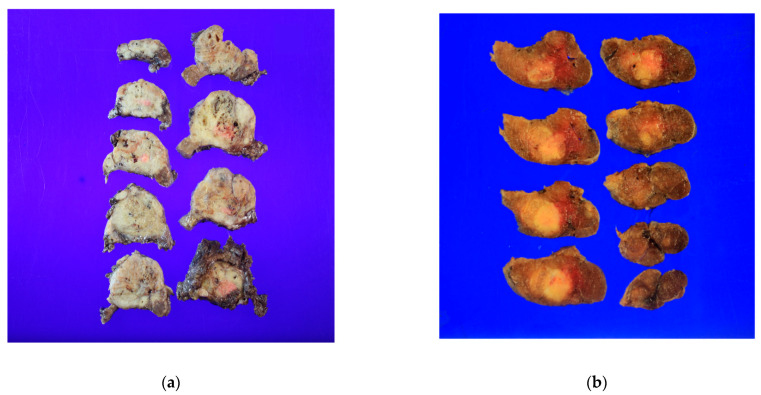
Gross picture of the specimen: (**a**) a specimen with gross extrathyroidal extension; (**b**) a specimen without gross extrathyroidal extension.

**Figure 2 cancers-14-04615-f002:**
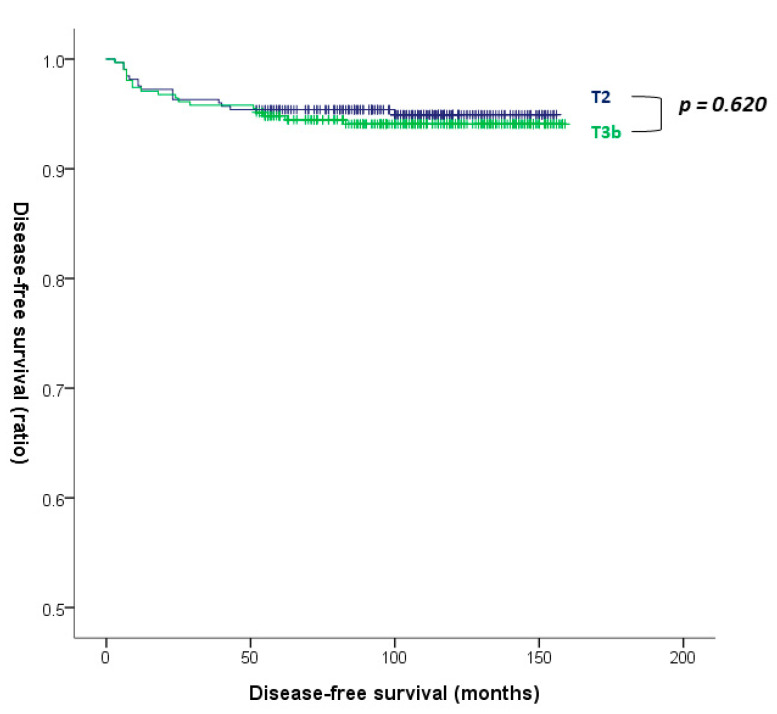
Disease-free survival curves of the T2 and T3b groups (*p* = 0.620 by log-rank test).

**Figure 3 cancers-14-04615-f003:**
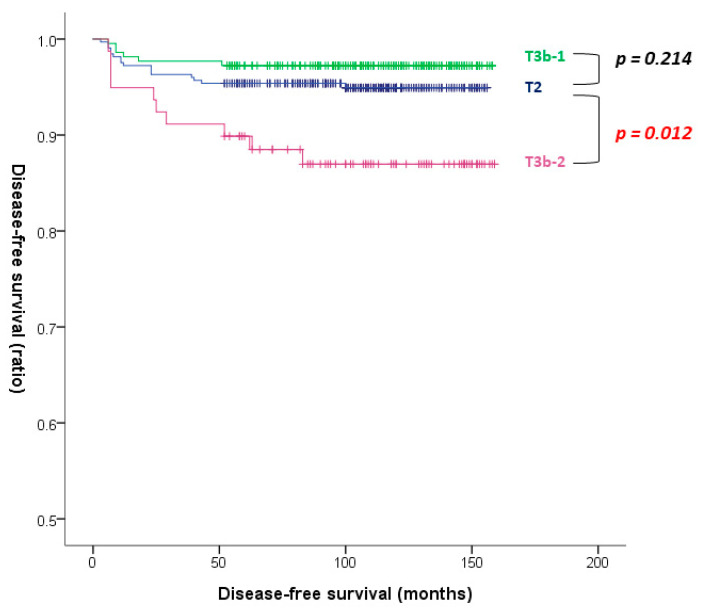
Disease-free survival curves of the T2, T3b-1, and T3b-2 disease (T2 vs. T3b-1: *p* = 0.214; T2 vs. T3b-2: *p* = 0.012 by log-rank test).

**Table 1 cancers-14-04615-t001:** Comparison of baseline clinicopathological characteristics between patients with T2 and T3b disease.

	T2 (*n* = 325)	T3b (*n* = 309)	*p*-Value
Age (years)	43.5 ± 14.6 (range, 12–83)	50.0 ± 13.6 (range, 11–83)	<0.001
Female	212 (65.2%)	252 (81.6%)	<0.001
Extent of surgery			<0.001
Lobectomy	184 (56.6%)	25 (8.1%)	
TT and/or mRND	141 (43.4%)	284 (91.9%)	
Tumor size (cm)	2.7 ± 0.5 (range, 2.1–4.0)	1.8 ± 1.0 (range, 0.3–6.5)	<0.001
Multifocality	125 (38.5%)	161 (52.1%)	0.001
Bilaterality	61 (18.8%)	125 (40.5%)	<0.001
Lymphatic invasion	134 (41.2%)	181 (58.6%)	<0.001
Vascular invasion	26 (8.0%)	23 (7.4%)	0.793
BRAF^V600E^ positivity	173/241 (71.8%)	217/252 (86.1%)	<0.001
Harvested LNs	17.5 ± 19.8	25.1 ± 26.3	<0.001
Positive LNs	4.3 ± 6.3	5.6 ± 6.9	0.013
N stage			<0.001
N0	138 (42.5%)	75 (24.3%)	
N1a	141 (43.4%)	141 (45.6%)	
N1b	46 (14.2%)	93 (30.1%)	
TNM stage			<0.001
Stage I	293 (90.2%)	188 (60.8%)	
Stage II	32 (9.8%)	121 (39.2%)	
Recurrence	16 (4.9%)	18 (5.8%)	0.614

Data are expressed as the patient’s number (%), or mean ± standard deviation. A statistically significant difference was defined as *p* < 0.05. Abbreviations: TT, total thyroidectomy; mRND, modified radical neck dissection; LN, lymph node; T, tumor; N, node; M, metastasis.

**Table 2 cancers-14-04615-t002:** Univariate and multivariate analyses of risk factors for recurrence in patients with T2 and T3b disease.

	Univariate		Multivariate	
	HR (95% CI)	*p*-Value	HR (95% CI)	*p*-Value
Age	0.958(0.934–0.982)	0.001	0.968(0.944–0.993)	0.012
Gender				
Female	ref.			
Male	2.830 (1.445–5.544)	0.002	2.506(1.277–4.918)	0.008
Tumor size	1.656 (1.206–2.275)	0.002	1.434(1.036–1.984)	0.030
Lymphatic invasion	6.129(2.372–15.833)	<0.001	4.822(1.851–12.562)	0.001
T stage				
T2	ref.			
T3b	1.185 (0.604–2.324)	0.621		
N stage		<0.001		
N0	ref.			
N1a	8.219 (1.927–35.053)	0.004		
N1b	8.592 (1.904–39.764)	0.005		

Data are expressed as hazard ratio (HR) and 95% confidence interval (CI). A statistically significant difference was defined as *p* < 0.05. Abbreviations: T, tumor; N, node.

**Table 3 cancers-14-04615-t003:** Comparison of baseline clinicopathological characteristics between patients with T2, T3b-1, and T3b-2 disease.

	T2 (*n* = 325)	T3b-1 (*n* = 217)	T3b-2 (*n* = 79)	*p*-Value (T2 vs. T3b-1)	*p*-Value (T2 vs. T3b-2)
Age (years)	43.5 ± 14.6 (range, 12–83)	49.7 ± 12.4 (range, 11–81)	51.4 ± 16.1 (range, 14–83)	<0.001	<0.001
Female	212 (65.2%)	183 (84.3%)	60 (75.9%)	<0.001	0.068
Extent of surgery				<0.001	<0.001
Lobectomy	184 (56.6%)	23 (10.6%)	1 (1.3%)		
TT and/or mRND	141 (43.4%)	194 (89.4%)	78 (98.7%)		
Tumor size (cm)	2.7 ± 0.5 (range, 2.1–4.0)	1.2 ± 0.4 (range, 0.3–2.0)	2.7 ± 0.5 (range, 2.2–4.0)	<0.001	0.147
Multifocality	125 (38.5%)	112 (51.6%)	43 (54.4%)	0.002	0.010
Bilaterality	61 (18.8%)	83 (38.2%)	36 (45.6%)	<0.001	<0.001
Lymphatic invasion	134 (41.2%)	112 (51.6%)	56 (70.9%)	0.017	<0.001
Vascular invasion	26 (8.0%)	13 (6.0%)	4 (5.1%)	0.375	0.372
BRAFV600E positivity	173/241 (71.8%)	151/178 (84.8%)	56/63 (88.9%)	0.002	0.005
Harvested LNs	17.5 ± 19.8	20.7 ± 24.2	31.8 ± 24.8	0.097	<0.001
Positive LNs	4.3 ± 6.3	4.3 ± 5.7	7.5 ± 7.3	0.992	0.007
N stage				0.002	<0.001
N0	138 (42.5%)	61 (28.1%)	13 (16.5%)		
N1a	141 (43.4%)	111 (51.2%)	29 (36.7%)		
N1b	46 (14.2%)	45 (20.7%)	37 (46.8%)		
TNM stage				<0.001	<0.001
Stage I	293 (90.2%)	135 (62.2%)	44 (55.7%)		
Stage II	32 (9.8%)	82 (37.8%)	35 (44.3%)		
Recurrence	16 (4.9%)	6 (2.8%)	10 (12.7%)	0.212	0.012

Data are expressed as the patient’s number (%), or mean ± standard deviation. A statistically significant difference was defined as *p* < 0.05. Abbreviations: TT, total thyroidectomy; mRND, modified radical neck dissection; LN, lymph node; T, tumor; N, node; M, metastasis.

**Table 4 cancers-14-04615-t004:** Univariate and multivariate analyses of risk factors for recurrence in patients with T2 and T3b-1 disease.

	Univariate		Multivariate	
	HR (95% CI)	*p*-Value	HR (95% CI)	*p*-Value
Age	0.938(0.907–0.969)	<0.001	0.955(0.923–0.988)	0.008
Gender				
Female	ref.			
Male	2.290 (0.989–5.301)	0.053		
Tumor size	1.626(1.000–2.646)	0.050		
Lymphatic invasion	5.599(1.895–16.544)	0.002		
T stage				
T2	ref.			
T3b-1	0.556 (0.218–1.422)	0.221		
N stage				
N0	ref.	0.001		
N1a	12.957 (1.718–97.707)	0.013		
N1b	11.026 (1.288–94.378)	0.028		

Data are expressed as hazard ratio (HR) and 95% confidence interval (CI). A statistically significant difference was defined as *p* < 0.05. Abbreviations: T, tumor; N, node.

**Table 5 cancers-14-04615-t005:** Univariate and multivariate analyses of risk factors for recurrence in patients with T2 and T3b-2 disease.

	Univariate		Multivariate	
	HR (95% CI)	*p*-Value	HR (95% CI)	*p*-Value
Age	0.957(0.930–0.985)	0.003	0.959(0.932–0.987)	0.004
Lymphatic invasion	4.961(1.871–13.158)	0.001	3.208(1.159–8.882)	0.025
T stage				
T2	ref.		ref	
T3b-2	2.640 (1.198–5.817)	0.016	2.659(1.159–6.099)	0.021
N stage				
N0	ref.	0.001		
N1a	6.965 (1.593–30.460)	0.010		
N1b	8.412 (1.817–38.933)	0.006		

Data are expressed as hazard ratio (HR) and 95% confidence interval (CI). A *p* value < 0.05 was considered statistically significant. Abbreviations: T, tumor; N, node.

## Data Availability

The data that support the findings of this study are available on request from the corresponding author. The data are not publicly available due to privacy or ethical restrictions.

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
