# Peer review of "Clinical Significance of Tumor Size in Gross Extrathyroidal Extension to Strap Muscles (T3b) in Papillary Thyroid Carcinoma: Comparison with T2"

_cancers, 2022, doi:10.3390/cancers14194615_

Round 1

Reviewer 1 Report

Dear Author,

The work entitled “Clinical Significance of Tumor Size in Gross Extrathyroidal Extension to Strap Muscles (T3b) in Papillary Thyroid Carcinoma: Comparison with T2” has been submitted to Cancers.

The work is well described, sustained by the collected data and shows relevant data.

Here you have some suggestions to improve your work:

-       Introduction could be better sustained with reference to past studies;

-       Methods should be more descriptive regarding the collection of data, source of the information, explaining if only pathologists reports were used; an effort to include some general data on PTC subtypes/ variants should be done;

-       Results could be easier to follow if tables are fused or less in their number; pictures of examples of cases with and without gross extrathyroid extension should be included;

-       Discussion is iterative, please consider to do not repeat observations.

Regards,

Author Response

We sincerely appreciate your comments that make our work improve further.

1)  Introduction could be better sustained with reference to past studies;

Response) I added more references and text was revised to better explain the purpose of this study and its causal relationship. Although many studies have focused on the effect of T3b itself, there are still few studies on the effect of tumor size in T3b. Developing from previous studies with only one size standard 4cm, in the present study, the cohort were divided into 2cm and 4cm using the existing T1 and T2 size categories. To the best of our knowledge, this is the first attempt of subcategories with these criteria and I added it to the introduction. Thank you again.

2) Methods should be more descriptive regarding the collection of data, source of the information, explaining if only pathologists reports were used; an effort to include some general data on PTC subtypes/ variants should be done;

Response) All information including tumor size, multifocality, bilaterality, lymphatic invasion, vascular invasion, BRAFV600E positivity, number of harvested lymph nodes and metastatic lymph nodes were confirmed by the pathologic reports and only T3b was confirmed intraoperatively by surgeons. I added this to ‘2.1 Patients’ and ‘2.2 TNM classification’ in Materials and Methods. As described in limitations, the PTC subtype/ variants were not included in our data. Further studies including these data are currently in progress, and we will draw more complete results in next study as your advice. Thank you.

3) Results could be easier to follow if tables are fused or less in their number; pictures of examples of cases with and without gross extrathyroid extension should be included;

Response) Following your comments, I merged table 3 and table 5 into table 3, which is visually more concise. 

The gross picture of specimen with and without gross extrathyroidal extension were attached to Figure 1, and added to text in the method

4) Discussion is iterative, please consider to do not repeat observations.

Response) Information already mentioned in the introduction or repeated results were deleted and corrected according to your advice. However, the first paragraph briefly summarized the results for the readers.

We thank you and the reviewers for the insightful comments. We believe that our manuscript has been improved as a direct result of the review process. We hope that the revised manuscript is now suitable for publication in CANCERS.

Sincerely,

Kwangsoon Kim, MD, PhD

Reviewer 2 Report

This is a review of outcomes for patients with T2 vs T3b PTC and is well written. The authors performed a novel analysis by segregating patients with T3b disease into two groups T3b-1 and T3b-2 based on tumor size and find that only those with larger tumors (T3b-2) had worse prognosis. This sheds light on previous observations in which the impact of T3b disease showed disparate findings. As such the manuscript is novel and has important findings for the management of patients with PTC.

Author Response

Reviewer #2

This is a review of outcomes for patients with T2 vs T3b PTC and is well written. The authors performed a novel analysis by segregating patients with T3b disease into two groups T3b-1 and T3b-2 based on tumor size and find that only those with larger tumors (T3b-2) had worse prognosis. This sheds light on previous observations in which the impact of T3b disease showed disparate findings. As such the manuscript is novel and has important findings for the management of patients with PTC.

Response) We sincerely appreciate your comments that make our work improve further. Some tables and figures have been modified to be more concise and repeated contents or expressions in dicussion have been corrected. Thank you again.

Reviewer 3 Report

Good paper and design for this study.

Describe clearly about when mRND (modified radical neck dissection) will be performed and how about the extension. Both extension of surgery (quite different in T2 and T3b in this study) and radioiodine therapy will affect the local recurrence. Show the data especially about radioiodine, and discuss about that.

If data available, show the recurrence rate between T1 and T3b-1

Author Response

1) Describe clearly about when mRND (modified radical neck dissection) will be performed and how about the extension.

Response) Modified radical neck dissection (mRND) was performed only when pathologic diagnosis of lateral cervical lymph node metastasis on preoperative biopsy. If it is suspected radiolofically but a preoperative biopsy was not performed, metastasis was confirmed through a frozen-section biopsy during surgery and the surgical extend was expanded to mRND if it was positive. Following your advice, in Materials and Methods, ‘2.3 Surgical treatment’ was added and decribed in detail.

2) Both extension of surgery (quite different in T2 and T3b in this study) and radioiodine therapy will affect the local recurrence. Show the data especially about radioiodine, and discuss about that.

Response) I appreciate for your points. In univariate analysis, extension of surgery had no effect on recurrence risk. In the tables of manuscript, only significant results are entered, so I attached a univariate & multivariate analysis that includes all variables in this letter. Unfortunately, however, data on the existenece of radioiodine therapy and its capacity were not included in the data collection process. This point was additionally written the limitations of our study, and in the further study including T1 and T3a, the completed results will be derived including radioiodine therapy. Thank you again.  

Table 2. Univariate and multivariate analyses of recurrence risk factors in patients with T2 and T3b

Univariate

Multivariate

HR (95% CI)

p-value

HR (95% CI)

p-value

Age

0.958(0.934-0.982)

0.001

0.968(0.944-0.993)

0.012

Gender

  Female

ref.

  Male

2.830 (1.445-5.544)

0.002

2.506(1.277-4.918)

0.008

Extent of surgery

  Lobectomy

ref.

  TT and/or mRND

1.016( 0.495-2.084)

0.966

Tumor size

1.656 (1.206-2.275)

0.002

1.434(1.036-1.984)

0.030

Multifocality

1.081(0.551-2.119)

0.821

Bilaterality

1.314(0.650-2.655)

0.447

Lymphatic invasion

6.129(2.372-15.833)

<0.001

4.822(1.851-12.562)

0.001

Vascular invasion

0.737(0.177-3.076)

0.676

BRAFV600E positivity

0.674(0.282-1.614)

0.376

T stage

  T2

ref.

  T3b

1.185 (0.604-2.324)

0.621

N stage

<0.001

  N0

ref.

Ref.

0.175

  N1a

8.219 (1.927-35.053)

0.004

3.588(0.728-17.672)

0.116

  N1b

8.592 (1.904-39.764)

0.005

2.260(0.416-12.275)

0.345

Data are expressed as hazard ratio (HR) and 95% confidence interval (CI). A statistically signifi-cant difference was defined as p < 0.05. Abbreviations: TT, total thyroidectomy; mRND, modified radical neck dissection; T, tumor; N, node..

Table 4. Univariate and multivariate analyses of recurrence risk factors in patients with T2 and T3b-1

Univariate

Multivariate

HR (95% CI)

p-value

HR (95% CI)

p-value

Age

0.938(0.907-0.969)

<0.001

0.955(0.923-0.988)

0.008

Gender

  Female

ref.

  Male

2.290 (0.989-5.301)

0.053

Extent of surgery

  Lobectomy

ref.

  TT and/or mRND

0.606(0.263-1.397)

0.240

Tumor size

1.626(1.000-2.646)

0.050

1.574(0.923-2.682)

0.096

Multifocality

0.476(0.186-1.217)

0.121

Bilaterality

0.428(0.127-1.446)

0.172

Lymphatic invasion

5.599(1.895-16.544)

0.002

2.653(0.839-8.385)

0.097

Vascular invasion

0.611(0.082-4.539)

0.630

BRAFV600E positivity

0.876(0.282-2.716)

0.818

T stage

  T2

ref.

  T3b

0.556 (0.218-1.422)

0.221

N stage

  N0

ref.

0.001

0.093

  N1a

12.957 (1.718-97.707)

0.013

6.581(0.778-55.638)

0.084

  N1b

11.026 (1.288-94.378)

0.028

4.332(0.441-42.599)

0.209

Data are expressed as hazard ratio (HR) and 95% confidence interval (CI). A statistically signifi-cant difference was defined as p < 0.05. Abbreviations: TT, total thyroidectomy; mRND, modified radical neck dissection; T, tumor; N, node..

Table 6. Univariate and multivariate analyses of recurrence risk factors in patients with T2 and T3b-2

Univariate

Multivariate

HR (95% CI)

p-value

HR (95% CI)

p-value

Age

0.957(0.930-0.985)

0.003

0.959(0.932-0.987)

0.004

Gender

  Female

ref.

  Male

1.541 (0.708-3.357)

0.276

Extent of surgery

  Lobectomy

ref.

  TT and/or mRND

1.146 (0.527-2.496)

0.731

Tumor size

1.218 (0.592-2.508)

0.593

Multifocality

1.207(0.558-2.609)

0.633

Bilaterality

1.700(0.758-3.814)

0.198

Lymphatic invasion

4.961(1.871-13.158)

0.001

3.208(1.159-8.882)

0.025

Vascular invasion

0.497(0.067-3.669)

0.497

BRAFV600E positivity

0.913(0.329-2.535)

0.861

T stage

  T2

ref.

ref

  T3b

2.640 (1.198-5.817)

0.016

2.659(1.159-6.099)

0.021

N stage

  N0

ref.

0.001

ref.

0.296

  N1a

6.965 (1.593-30.460)

0.010

3.079(0.611-15.523)

0.173

  N1b

8.412 (1.817-38.933)

0.006

2.051(0.337-12.488)

0.436

Data are expressed as hazard ratio (HR) and 95% confidence interval (CI). A statistically signifi-cant difference was defined as p < 0.05. Abbreviations: TT, total thyroidectomy; mRND, modified radical neck dissection; T, tumor; N, node.

3) If data available, show the recurrence rate between T1 and T3b-1

Response) Since this study was originally designed to compare T2 and T3b according to the tumor size,the cohort of this st\udy consists of T2, T3b-1, and T3b-2. According to your suggestion, the recurrence rates of T1 and T3b-1 from 2009 to 2017 were compared using Pearson’s chi-square test through the data of our institution.

Supplementary table 1. Comparison of recurrence rate between patients with T1 and T3b-1disease.

T1 (n=5280)

T3b-1 (n=217)

p-value

Recurrence

130(2.5%)

6(2.8%)

0.778

Supplementary table 2. Univariate Cox regression analyses of T stage as a risk factor for recurrence in patients with T1 and T3b-1

Univariate

HR (95% CI)

p-value

T stage

  T1

ref.

  T3b-1

1.118(0.493-2.533)

0.790

Currently, a further study on a large cohort including T1, T2, T3a and T3b for a longer period is in progress, and in a further study, analyses including comparison of clinicopathologic characteristics will be conducted.

We thank you and the reviewers for the insightful comments. We believe that our manuscript has been improved as a direct result of the review process. We hope that the revised manuscript is now suitable for publication in CANCERS.

Sincerely,

Kwangsoon Kim, MD, PhD
